# The Longevity Protein Klotho: A Promising Tool to Monitor Lifestyle Improvements

**DOI:** 10.3390/metabo13111157

**Published:** 2023-11-16

**Authors:** Jean-Sébastien Paquette, Caroline Rhéaume, Pierre Cordeau, Julie-Alexandra Moulin, Etienne Audet-Walsh, Virginie Blanchette, Jean-Philippe Drouin-Chartier, Alfred-Kodjo Toi, Angelo Tremblay

**Affiliations:** 1Primary Care Research and Innovation Laboratory (Laboratoire ARIMED), Groupe de Médecine de Famille Universitaire du Nord de Lanaudière, Centre Intégré de Santé et de Services Sociaux de Lanaudière, Joliette, QC J6E 5X7, Canada; 2Department of Family Medicine and Emergency Medicine, Faculty of Medicine, Laval University, Québec City, QC G1V 0A6, Canada; 3Vitam, Research Center on Sustainable Health, Université Laval, Québec City, QC G1V 0A6, Canada; 4Département de Médecine Moléculaire, Faculté de Médecine, Université Laval, Québec City, QC G1V 0A6, Canada; 5Department of Human Kinetics and Podiatric Medicine, Université du Québec à Trois-Rivières, Trois-Rivières, QC G1V 0A6, Canada; 6NUTRISS (Nutrition, Health and Society) Research Centre, Institute on Nutrition and Functional Foods (INAF), Laval University, Québec City, QC G1V 0A6, Canada; 7Faculty of Pharmacy, Laval University, Québec City, QC G1V 0A6, Canada; 8Department of Kinesiology, Faculty of Medicine, Université Laval, Quebec City, QC G1V 0A6, Canada; 9Centre de Recherche de l’Institut Universitaire de Cardiologie et de Pneumologie de Québec (IUCPQ), Quebec City, QC G1V 4G5, Canada

**Keywords:** lifestyle medicine, klotho, biomarker, aging, health

## Abstract

Aging is not a disease; it is a natural evolution of human physiology. Medical advances have extended our life expectancy, but chronic diseases and geriatric syndrome continue to affect the increasingly aging population. Yet modern medicine perpetuates an approach based on treatment rather than prevention and education. In order to help solve this ever-growing problem, a new discipline has emerged: lifestyle medicine. Nutrition, physical activity, stress management, restorative sleep, social connection, and avoidance of risky substances are the pillars on which lifestyle medicine is founded. The aim of this discipline is to increase healthspan and reduce the duration of morbidity by making changes to our lifestyle. In this review, we propose the use of klotho protein as a novel biomarker for lifestyle medicine in order to quantify and monitor the health status of individuals, as no integrative tool currently exists.

## 1. Introduction

As the global prevalence of chronic diseases continues to rise along with healthcare costs, it is critical to invest in their prevention and management. Worldwide, cancer, cardiovascular disease, diabetes, and chronic respiratory diseases killed around 33.2 million people in 2019, representing a 28% increase compared to 2000 [1]. In the United States, chronic diseases are the leading cause of death and disability and cost society USD 4.1 trillion annually [2]. In 2015, the United States ranked first in the world for health expenditure as a percent of gross domestic product but 31st in the world for life expectancy [3].

With the global population aging, the number of people with multiple chronic health conditions has been rising because medicine still focuses on treatment rather than prevention [3]. It is estimated that one in three people in the world lives with two or more chronic diseases [4]. One way to prevent and even reverse chronic diseases is through lifestyle changes through health promotion and education [4]. This also helps delay the onset of geriatric syndrome (frailty, cognitive decline, and reduced performance in the activities on the daily living scale) [5]. This is why a new discipline of medicine has emerged to specifically address this issue: lifestyle medicine [6]. The American College of Lifestyle Medicine (ACLM) promotes a science-based approach that integrates lifestyle factors to prevent and treat chronic conditions. There are six pillars of lifestyle medicine: nutrition, physical activity, stress management, restorative sleep, social connection, and avoidance of risky substances [6]. The goal of this discipline is not only to prolong the lifespan but to increase the healthspan by reducing the morbidity span [7]. A study estimated that adherence to four or five low-risk lifestyle factors (diet, physical activity, alcohol intake, etc.) at age 50 could extend life expectancy free of major chronic diseases (cancer, cardiovascular disease, or diabetes) by 7.6 years in men and 10.6 years in women when compared to people with no low-risk lifestyle factors [8].

Healthspan is defined as longevity without diseases and is often associated with a higher quality of life. Healthy longevity is a WHO (World Health Organization) priority [9]. Chronological age, the number of years a person has been alive, is a great predictor of disease prevalence and mortality risk but is unchangeable [10]. On the other hand, biological age is variable and measures the accumulation of physiological damage in individuals, meaning that two individuals of the same chronological age can have different biological ages [11]. Thus, a biological marker providing a quantifiable overall insight into the patient’s current health status would be of great use.

This review introduces the klotho protein as a potential novel, cost-effective biomarker and integrative tool to quantify and monitor the health status of individuals adopting lifestyle behavioral changes and summarizes current knowledge on the extent of klotho regulation across the six pillars of lifestyle medicine.

## 2. A Marker to Rule them All: Klotho, the Longevity Protein

### 2.1. Potential Healthspan Biomarker

Although many markers exist to monitor each of the six pillars of lifestyle medicine, biomarkers are mainly associated with a specific chronic disease or illness, such as glycated hemoglobin for diabetes and creatinine for renal failure, among others [12,13,14]. Unfortunately, when a healthcare professional wants to assess the lifestyle of a patient, the only reliable tools are questionnaires and connected devices [15]. In order to assess health rather than pathology, physicians take anthropometric measurements and prescribe blood tests to rule out diseases [16,17,18]. Thus, those tests are performed to establish a diagnosis, but no marker is routinely used to measure the patient’s healthspan or lifespan potential. However, a few longevity markers do currently exist, such as PhenoAge (algorithms to improve chronological age by adding 9 biomarkers found in routine blood tests) or GrimAge, which is an epigenetic clock that can evaluate the biological age of an individual using DNA methylation-based markers [11,19]. These tests are reliable for determining biological age, but there is little literature linking them to healthspan potential and even less to each of the pillars of lifestyle medicine. A new biomarker, the longevity protein klotho, might become a game-changing tool for measuring metabolic health and predicting the potential for healthy longevity.

### 2.2. Molecular Biology of Klotho

Discovered in 1997, the klotho gene (KL) codes for a transmembrane protein mainly expressed in the kidneys and, in smaller quantities, in the brain, parathyroid glands, peripheral blood cells, and vascular tissues [20,21]. The klotho protein has great potential to be used as a biological marker since a high serum level of klotho is associated with health and longevity [22].

Three homologs of KL have been described: α-klotho, β-klotho, and γ–klotho [22]. The α-klotho is secreted in the blood and has a pleiotropic effect on many physiological processes [23]. The β-klotho does not appear to have endocrine functions on its own and is predominantly expressed in the liver [23]. β-klotho might have potential diagnostic and therapeutic value in cardiovascular disease and psychiatric symptoms [24]. As for γ–klotho, its biological role remains elusive [25]. A study published in 2019 found that γ–klotho is significantly upregulated in triple negative breast cancer [26]. Because the reported studies primarily measured α-klotho, only α-klotho will be discussed in this article and will be referred to as klotho. The transmembrane klotho isoform forms a complex with fibroblast growth factor receptor 23 (FGF23) [27]. It can be cleaved by cell-surface protease (sheddases), thus creating a soluble circulating klotho [20]. It is released into the blood, cerebrospinal fluid, and urine and is an indicator of α-klotho gene expression [28,29,30].

This soluble form acts as a hormone and is associated with longevity [27]. As reviewed by Arroyo et al., KL expression decreases with age in mice and in humans and is associated with a shorter lifespan [21]. Research has shown that mice with a defective expression of the klotho gene aged twice as fast as those without the defect and displayed several characteristics of accelerated aging [31]. In mice, advanced aging is characterized by organ and skin atrophy, infertility, atherosclerosis, vascular endothelial dysfunctions, vascular calcification, cardiac hypertrophy, decreased bone mineral density, sarcopenia, glucose metabolism disorders, and impaired cognition [32]. Conversely, the overexpression of KL in mice extended their lifespan when compared to wild-type mice [33].

Since its discovery, several studies in humans have demonstrated the protective effect of klotho against various chronic diseases and conditions such as cancers, atherosclerotic heart disease, Alzheimer’s dementia, and physical decline [23,25,34]. Klotho inhibits at least four pathways that have been linked to aging: transforming growth factor β (TGF-β), insulin-like growth factor 1 (IGF-1), nuclear factor κB (NF-κB), and Wnt/β-catenin. Besides its role in the inflammatory processes, klotho also activates antioxidant pathways, such as nuclear factor-erythroid 2-related factor 2 (Nrf2) and forkhead transcription factors (FoxOs) [23]. FoxO plays an essential role in the insulin and insulin-like growth factor signaling pathway and thus also plays a role in cellular metabolism, aging, and longevity [35]. It was also recently discovered that kidney-secreted soluble klotho directly maintains hematopoietic stem cell pool size and differentiation and is involved in the regulation of bone marrow hematopoiesis by inhibiting inorganic phosphate uptake by hematopoietic stem cells [36].

### 2.3. Klotho and the Pillars of Lifestyle Medicine

Klotho levels are influenced by nonmodifiable factors, namely age and genetics, but interestingly, klotho can also be modulated by modifiable lifestyle factors [24]. Here, we explain in depth why the klotho protein could be a promising lifestyle biological marker used to monitor the effectiveness of a health program, given that the protein is directly involved in all of the six pillars of lifestyle medicine defined by the ACLM [37].

#### 2.3.1. Nutrition

The pillar of nutrition refers to the macronutrients and micronutrients we ingest daily and considers both the quantity and quality of diets [38]. ACLM and WHO recommend the Mediterranean diet as a gold standard diet. Since 1975, obesity has nearly tripled worldwide. An unhealthy diet can lead to obesity and poor clinical outcomes [1]. Health and weight are not always connected, but obesity can cause chronic medical conditions like type 2 diabetes, hypertension, atherosclerotic events, and other conditions [39]. In addition to the weight variable, clinicians use hip circumference to evaluate visceral fat, which can be a sign of metabolic syndrome [40]. Its mechanism is unknown, but it is characterized by a combination of glucose intolerance, lipid metabolism disorders, high blood pressure, and excessive weight, leading to type 2 diabetes and cardiovascular events [41]. This condition can also be assessed by using a bioimpedance scale or an abdominal scan such as a DXA scan [42,43]. These measurements can also be used in the assessment of obesity-related cardiovascular risks [44].

In order to evaluate the actual condition of the nutrition pillar, all the markers currently used serve mainly to identify dietary deficiencies (protein, albumin, vitamins, minerals (calcium, phosphorus, magnesium, and others)) or diseases (glucose for diabetes, cholesterol for dyslipidemia, and others). These markers are not very useful for assessing improvement or deterioration in the quality of foods consumed by a healthy individual within a lifestyle program or the effect of food quality on healthspan potential. Indeed, a healthy individual without pathology will have normal values for all these biological and physiological markers, and thus, any improvement in food intake quality will not be measurable.

From a more preventive and health-promoting perspective, the klotho longevity marker seems modulated by diet and could be a potential biomarker for the nutrition pillar. Recent studies have found that whole-diet interventions are effective in increasing klotho function. A low-calorie, high-protein diet increases klotho levels in a rat’s brain, while a phosphate-restricted diet improves klotho kidney expression in a mouse model of polycystic kidney disease [45]. In a cross-sectional cohort analysis of 8456 participants (NHANES), klotho plasma levels were compared to the Healthy Eating Index-2015 (HEI) in middle-to-older aged (40–79 years old) adults [46]. This index evaluates subjects’ compliance with US dietary recommendations by assessing the composition and content of daily food intake [47]. They found a positive correlation between the HEI scores and klotho protein levels, suggesting that adherence to healthy dietary patterns may benefit the prevention of aging and health maintenance. Another study compared the diets of 7906 subjects obtained from the same National Health and Nutrition Examination Survey 2007–2016 cohort [48]. They compared four diets, namely the Mediterranean adherence diet score (MDS), the low-carbohydrate diet score, a low-fat diet, and a low-carbohydrate diet, and their associations with klotho levels. They concluded that the MDS had a positive and significant association with klotho levels, while the three other diets showed no association. Moreover, the results of the MDS revealed that odds ratios for aging gradually decreased as the adherence score increased, indicating protective effects from aging in higher MDS groups. The MDS score is evaluated according to nine specific items (fruits, vegetables, cereal, fish, meat and alternatives, dairy products, alcohol, and olive oil) [48]. The Mediterranean diet focuses on minimally processed plant foods and adopts a food-based approach rather than a nutrient-based approach, which values the complex interactions between foods and their constituents. Restrictive diets such as low-fat and low-carbohydrate diets can be of poor quality and induce physiological stress through the excess or lack of certain macronutrients, whereas the Mediterranean diet is balanced and does not cause dietary stress [48]. However, another small-scale study, conducted in a group of 74 middle-aged (45–65 years old) sedentary adults, evaluated the diets of participants by questionnaire and found an increase in klotho levels with nut consumption and a negative relationship between adherence to the Mediterranean diet [49]. Although these results seem to contradict the results of the previous study, it is important to note that all associations observed in the study disappeared after controlling for lean mass index, suggesting that lean mass index is the main determinant of klotho levels due to the modulation of klotho expression by skeletal muscles. Furthermore, the results may not be extended to younger or physically active populations [49].

Other diet components can influence klotho levels. For example, curcumin, ginseng, and resveratrol have been found to induce klotho expression in animal models [23]. A study demonstrated curcumin-induced KL expression in a human proximal tubule epithelial cell line and in a mouse model via the suppression of the TGF-β signaling pathway [50]. Another study performed in humans (NHANES) demonstrated that the composite dietary antioxidant index (CDAI) was positively associated with klotho levels in a middle-aged population, mainly in nonsmoking males [51]. The CDAI is a summary score of multiple dietary antioxidants, such as vitamins A, C, and E, manganese, selenium, and zinc. Another vitamin that seems to have a positive effect on klotho levels is vitamin D. In fact, there are vitamin D-responsive elements near the mouse and human klotho genes. Vitamin D treatment induced the hyper-expression of klotho mRNA in different mouse renal cell lines [52]. Several preclinical studies demonstrated that exposure to vitamin D could modulate klotho activity [53,54,55]. A Japanese group that treated mice using a vitamin D-enriched diet found an increase in the expression of the klotho gene; the expression was upregulated and reached maximal levels 8 h after the administration of 1,25-dihydroxyvitamin D3 [56]. All things considered, we can safely say that a good and balanced diet is associated with higher levels of klotho, and this is consistent with the concept that a healthy diet improves longevity and healthspan.

#### 2.3.2. Physical Activity

Physical activity is the second pillar of lifestyle medicine and is an important element when improving lifestyle habits. Physical activity, as proven in many studies, reduces the risk of developing chronic diseases, improves quality of life, and increases lifespan and healthspan [7]. According to the WHO, more than a quarter of the world’s adult population (1.4 billion adults) are insufficiently active. In order to meet the global recommendations, a minimum of 150 min of moderate-intensity aerobic physical activity should be performed throughout the week. However, people tend to overestimate the duration and intensity of their exercise regimen and underestimate the time spent as sedentary beings [57]. When asked, 47–90% of older adults (60 to 92 years old) believe they meet this requirement. In reality, when using an unbiased measurement protocol, the percentage of older adults who meet the 150 min of exercise drops between 6–26% [57]. Several devices exist to monitor physical exercise, such as smartphones and watches [58]. For example, these provide information on the number of steps per day or kilometers walked, with 10,000 steps per day or 8 km being established as a goal for a healthy lifestyle [59]. Paired with physiological measures, such as the maximum rate of oxygen (VO_2_ max) or functional threshold power (FTP), these devices can be used to document the progress of a training session at the cardiovascular level [60]. Although exercise can influence the same biological markers as nutrition (blood glucose, lipid profile), these are not ideal markers for measuring the benefits of physical activity on healthspan potential.

In a mouse model, studies have shown that klotho levels increased after treadmill exercise in both young and old mice [61]. However, when physical activity is intense, klotho levels fall immediately after acute exercise and return to pre-exercise levels, or even higher, in the days that follow [62]. A 12-report meta-analysis of randomized and quasi-randomized controlled trials, comprising 621 participants aged 30 to 65 years old, has evaluated the effects of exercise on klotho protein and found that klotho levels consistently increase after exercise [63]. Different exercise programs were evaluated (aerobic vs. resistance), and they all had a beneficial effect on klotho expression, with the exception of intense exercise in a hot environment or done during military operational stress [63,64]. This can be explained by the fact that extreme physical activities can lead to excessive physiological stress [64]. Indeed, strenuous exercise induces an increase in free radicals and reactive oxygen species. This oxidative stress may lead to an increase in muscle damage, the amount of toxins, and, inevitably, cell death, which appears to halt the production of klotho [63]. In contrast, a transient increase in oxidative stress will generate aerobic-induced benefits by increasing the antioxidant system, including klotho [63]. It was also shown that individuals who exercised at three times the recommended intensity had a higher probability of developing subclinical coronary atherosclerosis [65]. On the other hand, light-to-moderate exercise has been shown to be superior to intense exercise in improving health and longevity. A study on Brazilian women aged over 60 showed a strong positive correlation between light physical activity and klotho levels [5]. Overall, this is consistent with the literature, which states that moderate physical activity improves life expectancy and longevity compared to a sedentary lifestyle [66]. Healthy exercise increases klotho levels, and this effect can then be used as a biomarker to evaluate the healthspan potential of any given individual.

#### 2.3.3. Stress Management

Despite its importance, this pillar of lifestyle medicine is all too often overlooked in clinical practice. It is known that stress is detrimental to health, can cause several chronic diseases, and reduce life expectancy [67]. Therefore, having a lifestyle medicine marker that can accurately quantify stress is essential for predicting healthspan. There are a few ways to measure stress through biological markers (cortisol) or questionnaires (GAD7: anxiety and PHQ-9: depression) [68,69].

Cortisol, the primary stress hormone, is increased in stressful situations, whether acute or chronic [70]. High cortisol concentration has a harmful effect on the entire physiology. In the brain, it will affect mood, behavior, cognition, and stress response programming [55]. In human studies, chronic stress in childhood and adolescence affects cognitive mechanisms and can lead to psychiatric disorders, depression, anxiety, and post-traumatic stress disorders [71]. Although there are few studies on this topic, it has been demonstrated that KL plays a critical role in the pathogenesis of major depression in a mouse model [72]. The authors found a decrease in the expression of KL in the nucleus accumbens (NAc), a region closely linked to stress and depression, in mice susceptible to chronic social defeat stress (CSDS) when compared to control mice [72]. They established that the modulation of KL via overexpression in the NAc has a behavioral effect on stress. They also demonstrated that the klotho present in the NAc can regulate the function of GluN2B, a N-methyl-D-aspartate receptor involved in learning and memory [72]. They concluded that high levels of klotho have an antidepressant effect in normal mice and that, in sensitive mice, it improves their behavioral responses to a stressful event [48]. Klotho was shown to be working in the opposite direction of cortisol and, likewise, was studied under chronic stress conditions in humans. A study evaluating klotho levels in military personnel under simulated military operational stress observed a decrease in klotho levels. From baseline through the end of the five-day protocol, circulating levels of klotho declined by 7.2% [64]. In contrast to the results of this study, Nakanishi et al. studied the use of klotho protein as a marker of stress in Japanese men and women aged 40 to 60 years old. They showed that klotho levels were significantly elevated in men who had poor stress management, while no correlations were found in women [73]. The same authors suggest that since klotho has anti-inflammatory effects, its increase might be a compensatory response to the stress felt by the subjects [74]. It is important to mention that the stress experienced by the Japanese participants was much lower than in the military personnel study and that the Japanese subjects were older. In a different study carried out with young maternal caregivers, klotho levels were lower in high-stressed women and were also correlated with higher depressive symptoms [75]. Remarkably, interventions, such as practicing yoga, have demonstrated an enormous potential as a stress reducer while increasing klotho transcript levels [76]. More in-depth studies are needed to better understand the mechanism underlying the influence of stress on klotho levels. Nonetheless, it can be said that stress influences klotho levels and is, therefore, a marker to monitor when assessing healthspan potential.

#### 2.3.4. Restorative Sleep

Sleep is another lifestyle pillar that has been extensively studied and has undeniable benefits on health and life quality. Sleep mediates many body functions, and unhealthy sleep behaviors have been linked to inflammatory responses, oxidative stress, and cardiovascular diseases [77]. Up to about 70 million Americans suffer from chronic sleep problems, according to the Center for Disease Control [78]. It has been established that for young and elderly people, the optimal sleep duration varies between 7 to 9 h per night [79]. As with the other pillars, biological markers and electronic devices are used to assess sleep quality. Electroencephalogram (EEG) values and melatonin are among the biomarkers used [80]. Questionnaires are also widely used to assess the presence of sleep pathology: the Sleep Hygiene Index, the Epworth Sleepiness Scale, the Mini-Sleep Questionnaire, and the Pittsburgh Sleep Quality Index, to name but a few [81]. Circadian rhythms follow a 24 h cycle regulated by light/dark alternations and feeding/fasting cycles [82]. Misalignment between the internal clock and the external environment, when persistent, can lead to circadian dysfunction. When the body can no longer cope, cortisol levels rise and cause insomnia, in addition to the multiple problems mentioned above [83].

There are a few studies on the effect of sleep on klotho protein concentration. The team of Huang et al., found a nonlinear association between sleep duration and klotho levels in the 13,765 participants of the NHANES database [79]. The results showed that klotho levels increased when sleep duration was around 5.5 h, whereas excessive sleep (>7.5 h) is associated with a significant decrease in klotho levels [79]. Although this was a cross-sectional study, the results are consistent with previous evidence that hypersomnia is detrimental to health and longevity [84]. In a second study, improved sleep quality in sedentary middle-aged adults was associated with higher klotho levels [85]. Although further studies are needed to fully ground klotho in this pillar, current data strongly suggest a positive correlation between good sleep habits and high klotho levels.

#### 2.3.5. Social Connection

Positive social interactions are a pillar of lifestyle medicine due to their important role in the well-being of individuals [37]. It is defined by the number, variety, and types of relationships a person has, such as having meaningful and regular social exchanges, a sense of support from friends, families, and others in the community, a sense of belonging, having close bonds with others, feeling loved, cared for, valued and appreciated by others, having more than one person to turn to for support (including emotional support when feeling down and physical support, like getting a ride to the doctor or grocery store or getting help with childcare on short notice), and access to safe public areas to gather (such as parks and recreation centers) [86,87,88,89,90]. Indeed, people who have good social relationships are happier, have fewer chronic diseases, and live longer [91]. This can be explained by the fact that historically, human beings have needed to rely on each other to survive. Humans are gregarious animals, and social isolation and loneliness are health risk factors [92]. Loneliness manifests itself when one’s social relationships are perceived as being less satisfying than what is desired [93].

As a general principle, all organisms must consume more energy than they spend to stay healthy. According to the social baseline theory, social connections are resources that help conserve energy and decrease the risk of predation and injury [94]. Our brain expects as a baseline to have social interactions [94]. Therefore, loneliness and social isolation are likely to be experienced as a cost to the body and to cause distress over time [91]. Social isolation and loneliness can cause cognitive decline and accelerate disease progression for people living with pre-existing health conditions [95]. A meta-analysis has shown that being socially connected increases the chance of survival by 50% [92]. When expected baseline requirement levels of social connection are not met, the state of alertness is increased in the central nervous system, activating the sympathetic system and the hypothalamic-pituitary axis. This activation can increase blood pressure, stress hormones (such as cortisol), and inflammatory responses, increasing the risk of chronic diseases [92]. Once again, the NAc, widely recognized as the center of reward and addiction, is also known, as previously shown in the stress management pillar, to play a role in social bonding, stress, and depression.

As components of social connection, sexual desire, and sexual function are also linked to overall health and are affected by an unhealthy lifestyle. A study found a strong correlation between klotho levels, sexual desire, and sexual function in sedentary middle-aged adults. The researchers concluded that chronological age did not predict the prevalence of sexual problems but that plasma klotho levels were a credible predictor of sexual status [96]. Further studies are needed to fully establish klotho in this pillar.

#### 2.3.6. Avoidance of Risky Substances

In lifestyle medicine, this pillar refers to toxic substances that can be consumed by a person [6]. Studies on substances such as tobacco, alcohol, and drugs are limited and contradictory. This indicates a lack of knowledge and that more in-depth studies are needed. However, we found that they all seem to have one thing in common: in the short term, klotho appears to generate a pro-inflammatory response, indicating that klotho may be produced to protect against damage produced by harmful substances.

##### Tobacco

The harmful effects of smoking on longevity, whether from cigarettes, drugs, or vapes, are well known. Each year, more than 7 million deaths are caused by tobacco use. Among active smokers, 80% are living in low- and middle-income countries [97]. Smoking can reduce life expectancy by at least 10 years and significantly increase the chances of developing chronic diseases such as atherosclerotic heart disease and cancer [97].

If klotho is a biomarker for healthspan potential, smoking would logically have a negative effect on its concentration in the blood. As a matter of fact, in a recent study, a decreased serum klotho level was measured in a group of smokers (*n* = 65) when compared to nonsmokers (*n* = 71) [98]. However, other studies have shown conflicting results. For example, one study observed a significant decrease in klotho serum levels in a group of 28 adults who went through a smoking cessation program for a period of 12 weeks [99]. Another study on healthy male smokers found that serum klotho levels were significantly higher in smokers than in nonsmokers [74]. Since smoking induces an inflammatory response, which involves the secretion of the pro-inflammatory cytokine interleukin (IL)-6, and klotho has been reported as an anti-inflammatory molecule, the authors hypothesized that the high levels of klotho observed in the study might be a compensatory response to inflammation-related stress caused by elevated IL-6 levels found in the smoking group. IL-6 promotes unhealthy aging phenotypes and a decreased likelihood of successful aging [100]. In older adults, elevated levels of IL-6 are involved in a phenomenon called “inflammaging”, which is defined as a chronic low-grade inflammatory state maintained by the age-associated increase in pro-inflammatory cytokines. IL-6 is also known as one of the common inflammatory biomarkers of frailty among other age-related diseases [101]. In this context, measuring IL-6 in parallel with klotho could help confirm that high klotho levels are indeed linked to an inflammatory response or state.

In a larger and more diverse group study, a nonlinear relationship between serum cotinine and klotho serum levels was demonstrated. Serum cotinine is a specific marker of smoking and reflects exposure to nicotine. The levels of klotho decrease as smoke exposure increases, then it reaches a tipping point where klotho levels start to increase [102]. Further research on the effect of cigarette use on klotho levels is needed.

##### Alcohol

The results published in the literature concerning alcohol are conflicting [103]. Some studies mention the beneficial effect of moderate alcohol consumption, as recommended by public health authorities, while others mention the harmful effects of alcohol on health, even at small doses [104]. Considering that alcohol abuse reduces longevity, it would be expected that klotho levels would decrease in individuals who consume excessive amounts of alcohol. However, in a study comparing cirrhotic and non-cirrhotic alcoholic patients, klotho levels were found to be significantly higher among cirrhotics. The authors suggest that inflammation induced by high levels of cytokines measured in cirrhotic patients could be the cause. Indeed, they found a direct correlation between tumor necrosis factor alpha (TNF-α) and klotho levels since TNF-α enhances the expression of a desintegrin and metalloprotease 10 (ADAM 10) [105]. ADAM10 is a sheddase that can cleave the klotho transmembrane protein, thus increasing klotho levels in the blood [103]. In contrast, a decrease in klotho levels was observed in a group of non-alcoholic, sedentary middle-aged adults as their weekly consumption of alcoholic drinks increased [29]. Dehydration, oxidative stress, and an increase in pro-inflammatory cytokines could all lead to lower klotho levels as alcohol consumption increases [29]. Although wine is considered a hallmark of the Mediterranean diet, contributing to its health benefits, it should be enjoyed responsibly and with moderation [29,106]. Further research on the effect of alcohol consumption on klotho levels is also needed.

##### Drugs

Few studies are available on the effect of illicit drugs on klotho expression levels in the blood. Cannabinoids are the most widely used drugs by people between the ages of 15 and 64 [107]. Some countries (The Netherlands and Canada) and US states (California, New York) have legalized the recreational use of cannabis. Therefore, it would be of interest to know the long-term effect of this recreational drug on the body and its effects on longevity. There are few studies on the effect of cannabis consumption on klotho protein, but none have been deemed very conclusive. One study found a negative correlation between klotho serum levels and age in a group of cannabis users when compared to a control group, suggesting that cannabis use has a detrimental effect on klotho [107].

Interestingly, in preclinical studies, the cannabinoid receptor is associated with improved longevity in *C. elegans* [108]. Trivedi et al. also observed an increase in klotho triggered by the cannabidiol (CBD) molecule [109]. CBD is not the same substance as “smoked” recreational cannabis, so the results do not apply to illicit drugs but rather to a cannabinoid molecule that is being studied for medicinal use [109]. Further research on the effect of cannabis use on klotho levels is needed.

#### 2.3.7. Holistic Integration of Klotho

There is a clear synergy between the pillars of lifestyle medicine. Our tool (Figure 1) is a modification of the PAVING Wheel that encourages people to pursue positive lifestyle changes by following different themes [110]. The PAVING STEPSS program focuses on 12 key topics; in contrast, our version focuses on the 6 pillars of lifestyle medicine. These two concepts demonstrate that lifestyle medicine is a holistic concept in which the six pillars or the 12 key topics are not isolated from one another but rather constitute different parts of a whole. This is why increasing klotho levels through nutrition and exercise will counteract the negative effects of stress. Smoking, on the other hand, undermines overall health by reducing exercise endurance and increasing the risk of anxiety-related behaviors [111,112]. The health gains achieved through good habits can be attenuated or even reversed by bad habits. Klotho has the potential to make it possible to study which pillars are the most critical or have the greatest effect on a healthy life expectancy.

As previously mentioned, in a non-diabetic person, blood glucose levels provide little information on health status or lifestyle modification. It is a marker of chronic disease but not of well-being or longevity. The same principle applies to anthropomorphic measurements. For a person who is not overweight or suffering from hypertension, weight and vital sign measurements do not help health professionals to ascertain whether that person’s lifestyle is improving. On the other hand, when properly interpreted in the patient’s clinical context, an increase in klotho levels clearly indicates a sustained positive adjustment following a lifestyle change. Higher levels of this protein are associated with increased longevity and can help prevent certain chronic diseases, including cancer, cardiovascular disease, and neurocognitive disorders. However, there seems to be an incongruity with respect to high levels of physical stress. Some studies suggest that klotho may also be increased in response to severe stress, such as over-exercise or liver diseases. This would appear to be a response to an inflammatory state. This is why, in order to interpret klotho correctly, it is essential to question the patient and have good knowledge of his or her medical condition. Given all the factors that can influence klotho, it is difficult to predict its expected level for a healthy individual. More studies, such as the one published in 2022 concerning a healthy cohort of men and women aged 18 to 89 years old, demonstrating the negative association between soluble klotho levels and age, should be performed [22]. Nevertheless, a comparison of consecutive klotho measurements can enable rapid assessment of a lifestyle change.

Based on the literature, we propose a classification model, including four klotho-level categories (Figure 2). In the first group, defined as “klotho deficiency”, klotho levels are below the average expected for a healthy individual and are associated with reduced longevity and increased risk of chronic disease. A lower serum concentration of klotho was proven to be a marker of increased mortality rates in American adults with lower levels of physical activity [113]. In an aging group, lower klotho levels were also associated with worse cognition, frailty, higher dependence on activities for daily living, and falls [114]. In the second group, referred to as “normal klotho”, klotho levels are associated with normal longevity in relation to the subject’s age. In the third group, called the “healthy group”, lifestyle choices are optimized, and the klotho levels are above the normal value, which is associated with increased longevity and better general health. In the last group, identified as “inflammatory klotho”, increases in klotho levels are associated with a response to stress or inflammation. Further work is needed to better understand this increase in klotho levels, which appears to be a compensatory physiological inflammatory response.

Most of the studies we found are observational (Table 1). As we have shown, klotho has an important role to play in four of the six pillars of lifestyle medicine (nutrition, physical activity, stress management, and restorative sleep). Further studies are needed on the pillars of social connection and avoidance of risky substances. Nonetheless, randomized trials would be mandatory to demonstrate its use as a biomarker in all pillars of lifestyle medicine. Punctually measuring blood klotho concentrations can provide valuable information, but when assessed regularly, it can become a powerful tool since it is the variation of klotho levels over time in one individual that allows extrapolation of its general health path.

## 3. Conclusions

Based on this narrative analysis, klotho is a very promising marker candidate for lifestyle medicine due to its potential involvement in the six pillars of lifestyle medicine. Although we have identified knowledge gaps that warrant further study (randomized trials) to better understand the use of klotho in monitoring the effect of a lifestyle change intervention, it has enormous potential to enable objective, quantitative, and rapid monitoring of the overall health and the healthspan of patients. Klotho could be used as a marker in clinical studies where it is difficult to control the entire patient environment. Klotho is easy to quantify and, in the case of age-related diseases, would be an excellent marker to follow, as some diseases show no perceptible symptoms for a long period of time.

The world’s population is growing, aging, and becoming increasingly sedentary. A healthcare system that focuses solely on treatment is unsustainable, and if this trend continues, healthcare costs will be astronomical as the population’s health declines. The recent pandemic has shown us the capabilities of modern science and the importance of being healthy. It has also clearly demonstrated the extent of social inequalities on a global scale. A 2019 report by The Institute of Health Metrics and Evaluation analyzed data from more than 190 countries and found that what people eat and fail to eat is the leading cause of disease and death [37]. Health promotion and prevention, coupled with simple, cost-effective biomarkers, will become a necessity for tomorrow’s medicine. Lifestyle medicine and allopathic medicine are not mutually exclusive but rather work together in the patient’s best interest. We must invest in health today to reap the benefits later, but first, efforts need to be focused where they are needed most so that health becomes accessible to all. Health is a multifactorial concept, and as long as there are low-income areas, low levels of education, and food-insecure households, inequalities will persist, as will chronic disease.

## Figures and Tables

**Figure 1 metabolites-13-01157-f001:**
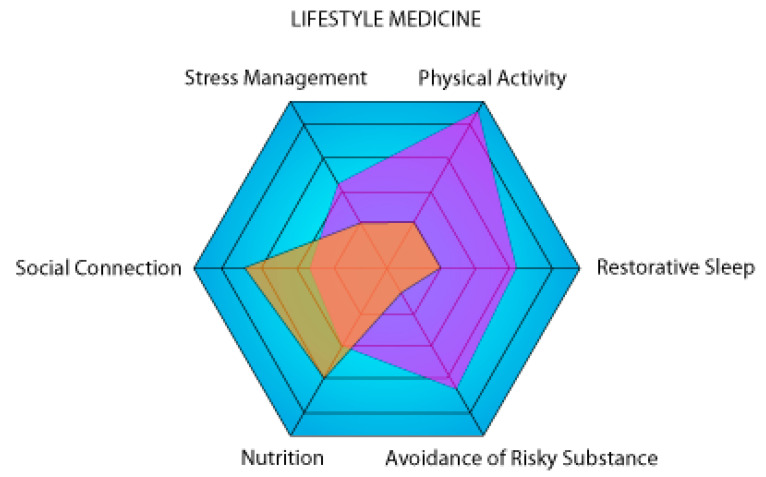
Graphical representation of two theoretical individuals (violet and orange) on the six axes of lifestyle medicine. The higher a person’s surface area on the representation, the lower the risk of developing a chronic disease. In this example, the purple person is a high performer focusing on physical activity, while the orange person is sedentary with a lot of social connections.

**Figure 2 metabolites-13-01157-f002:**
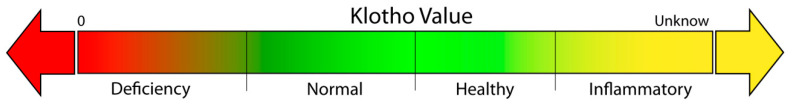
Representation of the four proposed categories of klotho values based on an individual’s baseline situation. Klotho deficiency is associated with a lower healthspan; increased klotho is associated with a higher healthspan, and an inflammatory klotho value indicates a possible stress response.

**Table 1 metabolites-13-01157-t001:** Effect of lifestyle on klotho serology.

Lifestyle Pillar	Effect on Klotho	Type of Studies	References
**Nutrition**			
Mediterranean diet Low-carb diet Low fat diet Healthy Eating Index	Increase No effect No effect Increase	Observational Observational Observational Observational	[48,49] [48] [48] [46]
**Physical activity**			
Low intensity High intensity	Increase Decrease	Observational, RCT RCT	[5,61,63] [62,64]
**Stress Management**			
Acute stress Chronic stress Yoga	Decrease Increase/Decrease Increase	Observational Observational RCT	[64] [73,75] [76]
**Restorative sleep**			
5.5–7.5 h ≤ 5.5–≥ 7.5 h	Increase Decrease	Observational, RCT Observational	[79,85] [79]
**Social connection**			
Sexual	Increase	Observational	[96]
**Risky substance**			
Tobacco Alcohol Drug	Increase/Decrease Decrease/Increase Decrease/Increase	Observational, RCT Observational, Literature survey Observational, Interventional	[74,98,99] [29,105] [107,108,109]

RCT, randomized controlled trial.

## Data Availability

The data presented in this study are available within the article.

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
