# Peer review of "The Longevity Protein Klotho: A Promising Tool to Monitor Lifestyle Improvements"

_metabolites, 2023, doi:10.3390/metabo13111157_

Round 1

Reviewer 1 Report

Comments and Suggestions for Authors

This review summarizes the current findings on the protein klotho as a potential biomarker for longevity. The article stated some key points with adequate literature cited. Some issues need to be addressed before consideration for publication. 

Comments:

1. The introduction section needs an overview of this review, including the objectives, importance of the topic, what's new about the knowledge, etc. 

2. Section 2.1. Here a more detailed explanation may be needed for "phenoage""Grimage""chronological age" and "biological age". There are more and more statements and definitions on these terms, so a clarification would be helpful.

3. Does klotho work similar as surtuins? Or do they monitor similar biological pathways regarding their physiological effects? The molecular biology of klotho reminds me of surtuins. 

4. Since klotho is secreted in blood. Are there any studies indicating it has beneficial or maintainance role in the hematopoietuc system?

5. For sections 2.3.1 and 2.3.2, the authors summarized some studies on how healthy dietary patterns and physical activity improve klotho levels. Do these studies also report beneficial physiological effects brought by the increased klotho levels? This should also be stated and discussed to link increased klotho levels by these factors with a improved healthspan. 

6. "Some studies suggest that klotho may also be increased in response to severe stress, such as over-exercise or liver diseases." In this case, how can klotho be served as an indicator for longevity?

7. The article proposed four categories of klotho level for prediction of health status. So what are the values? Based on what did the authors reach such conclusion? There will need to be supporting literature with values. 

Author Response

1. Summary

Thank you very much for taking the time to review this manuscript. Please find the detailed responses below and the corresponding revisions and corrections in track changes in the re-submitted file.

2. Point-by-point response to Comments and Suggestions for Authors

Comment 1: The introduction section needs an overview of this review, including the objectives, importance of the topic, what's new about the knowledge, etc.

Response 1: Thank you for pointing this out. We agree with this comment. Therefore, on page 2, under section 1., we have move the first paragraph of 2.1 section to introduction (3rd paragraph added) and have added a 4rd paragraph to provide a brief overview of the review.

Comment 2: Section 2.1. Here a more detailed explanation may be needed for "phenoage""Grimage""chronological age" and "biological age". There are more and more statements and definitions on these terms, so a clarification would be helpful.

Response 2: Agree. We have, accordingly, added brief definitions for these terms on page 2, under section 2.1, 1st and 2nd paragraphs, to clarify this point.

Comment 3: Does klotho work similar as surtuins? Or do they monitor similar biological pathways regarding their physiological effects? The molecular biology of klotho reminds me of surtuins.

Response 3: Thank you for this relevant and interesting comment. Sirtuins are class III histone deacetylases that comprise seven members and are widely expressed in mammals. They are involved in a myriad of cellular processes, including apoptosis, proliferation, differentiation, epithelial-mesenchymal transition, aging, DNA repair, senescence, viability, survival, and stress response (Onyiba et al., 2022). In the literature, there are many controversial reports surrounding the seven members of the sirtuin family (Weiwei, 2014). In addition, theories regarding the usefulness of sirtuins as longevity enzymes are ambiguous (Ziętara et al. 2023). Nonetheless, a recent review has listed the interactions between klotho and sirtuins at different levels of gene expression and signaling pathways (Rostamzadeh et al., 2023). In conclusion, sirtuins could certainly become a therapeutic target in the future, but further studies are needed before sirtuins can be considered as biomarkers of lifespan in the six pillars of lifestyle medicine. Therefore, this article focuses solely on klotho.

Comment 4: Since klotho is secreted in blood. Are there any studies indicating it has beneficial or maintainance role in the hematopoietuc system?

Response 4: Thank you for this very relevant question. Indeed, the role of klotho in the maintenance of hematopoietic stem cells and its involvement in the regulation of bone marrow hematopoiesis were recently identified by Du et al. (2022). Therefore, we have added this information on page 3, under section 2.2., 4th paragraph.

Comment 5: For sections 2.3.1 and 2.3.2, the authors summarized some studies on how healthy dietary patterns and physical activity improve klotho levels. Do these studies also report beneficial physiological effects brought by the increased klotho levels? This should also be stated and discussed to link increased klotho levels by these factors with an improved healthspan.

Response 5: Thank you for this question. The studies summarized in sections 2.3.1 and 2.3.2 do not report direct beneficial physiological effects brought by the increased klotho levels due to their retrospective design. However, in the study performed by Wu et al. (2022), odds ratios (ORs) for aging were calculated and the results of the Mediterranean adherence diet score (MDS) revealed that ORs gradually decreased as the adherence score increased, indicating protective effects from aging in higher MDS groups. Therefore, we have added this information on page 4, under section 2.3.1., 3rd paragraph.

Comment 6: "Some studies suggest that klotho may also be increased in response to severe stress, such as over-exercise or liver diseases." In this case, how can klotho be served as an indicator for longevity?

Response 6: Thank you for this comment. Indeed, klotho can also be modulated as a response to stress, which we consider to be “inflammatory klotho” (as illustrated in Figure 2). Since klotho can be influenced by the 6 pillars of lifestyle medicine, its measurement would provide a holistic vision of the patient's metabolic state. It is then important to separate inflammatory situations from the effects of improving lifestyle habits on klotho levels. To answer this question, there are two important elements to consider: 1) the health status of the patient to verify the presence or not of an underlying inflammatory disease and 2) the timing of the blood test to ensure that the patient is not in a transient inflammatory state (after strenuous exercise, for example), two situations that could interfere with the results. This is why, as mentioned in section 2.3.7., it is important to interpret klotho levels within the global clinical context of the patient and its use must be longitudinal (before and after an intervention). More precisely, klotho levels would be measured before engaging the patient in a lifestyle intervention, as a baseline, and at different point of the intervention (e.g., after 1, 3 or 6 months, etc.). This would allow verifying the effect of the patient's adherence to the intervention, with the aim of bringing their klotho levels to a healthy level and therefore improving their healthspan.

Comment 7: The article proposed four categories of klotho level for prediction of health status. So what are the values? Based on what did the authors reach such conclusion? There will need to be supporting literature with values.

Response 7: According to our literature review, we have proposed a model that integrates 4 categories of klotho levels. This model is purely theoretical, based on the integration of all of the observations in the literature review, in order to better represent the modulation of klotho according to the different contexts in which the protein is measured. Since there is currently a lack of data in the literature to obtain these types of values (the exact value of klotho in each categories), we have refrained from providing precise values in the article. Nonetheless, two studies (Kresovich, 2021 and Espuch-Oliver, 2022) have given us an idea of the order of magnitude of the klotho levels in serum. Kresovich divided a group of 10 069 people aged from 40-79 years in 4 quartiles based on mortality rate. Participants in the lowest quartile (< 666 pg/mL) had worse survival rates to those with higher concentration. The 4 quartiles were 1) < 666 pg/mL, 2) 666-808 pg/mL, 3) 809-985 pg/mL and 4) > 985 pg/mL. In the second article, the author quantified klotho expression in a population of men and women aged 18 to 85. The groups were divided into 3 sub-groups based on age: 18-34 years old: 932.6 ± 575.6 pg/mL, 35-54.9 years old: 796.7 ± 317.4 pg/mL and 55-85 years old: 612.1 ± 198.2 pg/mL. Both studies were performed in healthy populations. This is why, in the future, we should extend the quantification of klotho to unhealthy populations in order to get a more accurate picture of the situation.

3. Response to Comments on the Quality of English Language

N/A

4. Additional clarifications

N/A

Reviewer 2 Report

Comments and Suggestions for Authors

Paquette et al. performed a very interesting review study to explore the potential implication of klotho in monitoring lifestyle improvement. This manuscript, while providing valuable data, has several concerns that need to be addressed before being considered for publishing.

1.     It would be better to add the specific statistical results of the studies in Table 1.  

Reviewer 3 Report

Comments and Suggestions for Authors

This manuscript nicely reviewed studies relating to klotho and its potential uses as a biomarker to assess and monitor the six pillars of lifestyle medicine, including nutrition, physical activity, stress management, restorative sleep, social connection, and avoidance of risky substances. I only have some very minor suggestions as detailed below.

In the “Nutrition” section, the authors indicated that the associations with the Mediterranean diet “disappeared after controlling for lean mass index.” Kindly elaborate on this. Does the lean mass index itself correlate to the klotho level or the Mediterranean diet?

As mentioned in the “Tobacco” section, IL-6 shows a correlation with successful aging. That makes me wonder about other biomarkers, such as IL-6, under investigation for aging and longevity. Some quick discussion and comparison with klotho here would be really appreciated.

In addition, it seems a little early to me to make the conclusion stating “[k]lotho makes it possible to study which pillars are the most critical or have the greatest effect on a healthy life expectancy” since uses of klotho as a biomarker for a healthy life expectancy require further validation itself.

Although they have been mentioned in the text, it is suggested to add the corresponding references in Table 1 in case the readers would like to read more about those studies.

English editing comments:

(1)      The word “improvemen” in the title seems meant to be improvements.

(2)     Kindly correct the numbering for the “Restorative sleep,” “Social connection,” “Avoidance of risky substances” section and their subsections as well as for the “Holistic integration of klotho” subsection.

(3)      There are two “.” after the second citation of [70].

(4)      A space is missing in the “Alcohol” subsection between “patients” and “could.”

Comments on the Quality of English Language

Comments are provided in the main comments to authors.  

Round 2

Reviewer 1 Report

Comments and Suggestions for Authors

This review provides  a comprehensive summary on protein klotho as a potential biomarker to quantify and monitor people's health status. Updated literature are included and the authors have successfully addressed all the previous comments. No additional comments from my end. Therefore, the manuscript in this form is recommended for publication.